# A Cross-Sectional Survey of Oral Adverse Events and Oral Management Needs in Outpatients Receiving Cancer Drug Therapy

**DOI:** 10.3390/cancers17040641

**Published:** 2025-02-14

**Authors:** Yuki Sakai, Kouji Katsura, Masaaki Kotake, Akira Toyama

**Affiliations:** 1Department of Pharmacy, Niigata University Medical and Dental Hospital, Niigata 951-8520, Japan; masaakikota.fs4@nuh.niigata-u.ac.jp (M.K.); toyama@med.niigata-u.ac.jp (A.T.); 2Department of Outpatient Cancer Chemotherapy Center, Niigata University Medical and Dental Hospital, Niigata 951-8520, Japan; 3Department of Oral Radiology, Niigata University Medical and Dental Hospital, Niigata 951-8520, Japan; katsu@dent.niigata-u.ac.jp

**Keywords:** outpatient, cancer drug therapy, oral adverse events, oral management

## Abstract

Oral adverse events during cancer drug therapy cause severe pain, difficulty eating, loss of taste, and impaired speech; they may also affect patient quality of life. We investigated the incidence and severity of oral adverse events in outpatients receiving cancer drug therapy and the necessity for intervention by medical professionals. The results of our questionnaire-based survey suggest that oral adverse events negatively affect the quality of life of outpatients receiving cancer drug therapy. More than a quarter of outpatients receiving cancer drug therapy indicated the need for oral management through dental interventions. Patients receiving 5FU-, taxane-, and anthracycline-based regimens, in particular, may require oral management by medical professionals.

## 1. Introduction

Oral adverse events during cancer drug therapy not only cause severe pain, difficulty eating, loss of taste, and impaired speech but may also affect the quality of life of patients [1]. Several scientific societies have developed guidelines for the oral management of patients receiving cancer treatment [1,2,3,4]. The Multinational Association of Supportive Care in Cancer/International Society of Oral Oncology (MASCC/ISOO) emphasize the importance of appropriate oral management by multidisciplinary medical teams involving well-trained dentists or dental hygienists to prevent and treat oral adverse events [3,4]. Oral management is generally performed by multidisciplinary teams in patients receiving bone marrow transplantation and head and neck radiotherapy because the incidence of oral adverse events observed during these treatments is high [5,6,7]. In contrast, outpatients receiving cancer drug therapy are often overlooked due to the assumption that the incidence of oral adverse events is low [1]. This group may thus not receive appropriate oral management. However, a few previous studies suggest that this assumption may not hold true [8,9], as more aggressive cancer drug regimens are increasingly being administered on an outpatient basis. This shift raises concerns regarding the adequacy of oral management in this patient population.

Therefore, we hypothesized that oral adverse events are more prevalent and severe among outpatients receiving cancer drug therapy than previously assumed, necessitating a reassessment of current oral management practices for these patients. While existing guidelines emphasize the importance of oral management, they predominantly address inpatients, and evidence regarding the burden of oral adverse events in outpatients remains limited. This study aimed to bridge this gap by examining the reported data on the frequency and severity of oral adverse events in outpatients receiving cancer drug therapy. Further objectives were to clarify patient needs for oral management, and to identify drugs that affect oral adverse events, facilitating the development of tailored management strategies for this growing patient population.

## 2. Materials and Methods


**Patients**


This cross-sectional survey was conducted at the Outpatient Cancer Chemotherapy Center of Niigata University Medical and Dental Hospital from 1 to 30 September 2022. Patients (1) with cancer, (2) who were receiving cancer drug therapy at the Outpatient Cancer Chemotherapy Center, and (3) who were over 18 years of age were eligible for inclusion. Patients (1) who did not agree to participate in the study, (2) who were receiving or had received radiation therapy, and (3) who were receiving oral care from dental professionals such as dentists or dental hygienists were excluded. This study was approved by the Ethics Committee of the Niigata University School of Medicine (approval number: 2022-0120).


**Data collection and survey**


This survey was conducted using a questionnaire designed with reference to the Patient-Reported Outcome (PRO) Common Terminology Criteria for Adverse Events (PRO-CTCAE) (Table 1) [10,11]. Data on the incidence and severity of oral adverse events, the need for oral adverse event management by dental professionals, and the effect of these events on quality of life were collected and evaluated.

In addition, we reviewed patient medical records to collect detailed information on sex, age, cancer type, and cancer drug therapy. Specifically, we recorded whether patients received 5-fluorouracil(5FU)-based (e.g., 5-fluorouracil, capecitabine), taxane-based (e.g., paclitaxel (PTX), docetaxel (DTX)), anthracycline-based (e.g., doxorubicin (DXR), epirubicin), or other regimens. This classification was based on the Clinical Guidance of Management for Mucositis, which categorizes agents with a high risk of inducing oral mucositis [1].


**Statistical analysis**


The sample size was not set because this was a cross-sectional study. The relationship between patient characteristics and major oral adverse events was analyzed using EZR ver.1.66 (Saitama Medical Center, Jichi Medical University, Saitama, Japan) [12], which is a graphical user interface for R (The R Foundation for Statistical Computing, Vienna, Austria). It is a modified version of the R commander designed to add statistical functions frequently used in biostatistics. Uni- and multivariate logistic regression analyses were used to identify the predictors of major oral adverse events. *p* < 0.05 was considered statistically significant.

## 3. Results

Two hundred and sixteen patients were enrolled in this study. The patient characteristics are shown in Table 2.

More than half of the patients were middle-aged (aged 40–69 years). The major cancer types were breast, lung, and gastrointestinal cancers. Patients had received the following cancer drug therapies: ICIs (e.g., nivolumab (480 mg/body,day1, every 28 days), pembrolizumab (400 mg/body,day1, every 42 days)), taxane-based regimens (e.g., cetuximab + PTX (250 mg/m^2^ and 80 mg/m^2^,day1 every 7 days), DTX (60–75 mg/ m^2^,day1, every 21 days), DTX + cyclophosphamide (CPA) (75 mg/m^2^ and 600 mg/m^2^,day1, every 21 days), PTX (80 mg/m^2^,day1, every 7 days), carboplatin + nab-PTX (AUC6 day1 and 100 mg/m^2^,day1, 8, 15, every 21 days), nab-PTX + gemcitabine (GEM) (125 mg/m^2^ and 1000 mg/m^2^, day1, 8, 15, every 28 days)), 5FU-based regimens (e.g., 5FU + oxaliplatin (L-OHP) (400/2400 mg/m^2^ and 85 mg/m^2^,day1, every 14 days), 5FU + irinotecan (CPT-11) (400/2400 mg/m^2^ and 150 mg/m^2^,day1, every 14 days), L-OHP + capecitabine (130 mg/m^2^,day1 and 2000 mg/m^2^,day1–14, every 21 days)), anthracycline-based regimens (e.g., DXR+CPA (60 mg/m^2^ and 600 mg/m^2^,day1, every 21 days), rituximab + DXR + vincristine + CPA + prednisolone (375 mg/m^2^, 50 mg/m^2^, 1.4 mg/m^2^, 750 mg/m^2^, day1 and 60 mg/m^2^ day1–5, every 21 days)), cisplatin (CDDP)-based regimens (e.g., CDDP + GEM (25 mg/m^2^ and 1000 mg/m^2^,day1, 8, every 21 days)), and monoclonal antibody drugs targeting epidermal growth factor receptor (EGFR) (e.g., panitumumab (6 mg/kg,day1, every 14 days). The most commonly used drug therapies were ICIs, and taxane- and 5FU-based regimens (31.0, 22.7, and 15.3%, respectively).


**Incidence of oral adverse events**


One hundred and twenty-seven patients (58.8%) experienced adverse oral events (Table 3). These included dysgeusia, oral mucositis, xerostomia, tongue coating, stickiness, oral discomfort, oral pain, and numbness around the mouth (Figure 1). The most prevalent oral adverse events were dysgeusia, oral mucositis, and xerostomia, with incidence rates of 25.9, 21.3, and 19.4%, respectively (Table 4).


**Patient needs for management of oral adverse events by medical professionals**


Sixty-eight patients (53.5%) wanted improvement of their oral adverse events. Furthermore, 44 patients (34.6%) had oral adverse events that affected their quality of life, and 34 patients (26.8%) wanted professional oral care from well-trained dentists and dental hygienists. Thirty-two patients (25.2%) experienced oral adverse events that were as severe as or even more severe than other adverse events (Table 5).


**The relationship between patient characteristics and major oral adverse events**


The relationship between oral adverse events and patient characteristics is shown in Table 3. Dysgeusia, oral mucositis, and xerostomia were the major oral symptoms (Figure 1).

Patients with breast cancer experienced significantly more oral adverse events than other patients (OR = 6.38; 95% CI = 1.05–38.90; *p* = 0.04). Patients receiving 5FU- and taxane-based regimens had significantly more oral adverse events than those receiving other regimens (OR = 9.37; 95% CI = 1.74–50.40; *p* < 0.01 and OR = 5.63; 95% CI = 1.91–16.60; *p* < 0.01, respectively). The incidence of oral mucositis did not differ significantly between any of the regimens. Dysgeusia was significantly more common in the taxane-based regimens than in the other regimens (OR = 4.76; 95% CI = 1.61–14.10; *p* < 0.01). The incidence of xerostomia was significantly higher in the 5FU-based regimens than in the other regimens (OR = 16.5; 95% CI = 1.74–157.00; *p* = 0.01) (Table 4).

## 4. Discussion

The objective of this study was to clarify the frequency and severity of oral adverse events in outpatients receiving cancer drug therapy and to clarify patients’ needs for oral management by medical professionals. Our findings have several important clinical implications. First, more than half of the outpatients receiving cancer drug therapy had oral adverse events. Second, more than a quarter of the patients who experienced oral adverse events felt that these symptoms were as severe as or even more severe than other non-oral adverse events. Third, more than a quarter of the patients who experienced oral adverse events wanted oral management through professional intervention by well-trained dentists and dental hygienists. These results indicate that oral adverse events have a significant negative impact on patients receiving outpatient cancer drug therapy.

The guidelines for supportive care in cancer have described the importance of interventions by multidisciplinary medical teams involving well-trained dentists and dental hygienists to manage oral adverse events [3,4,13]. The guidelines indicate that professional dental intervention is essential for patients at high risk of oral adverse events, such as those undergoing bone marrow transplantation and head and neck radiotherapy. Conversely, outpatient chemotherapy is currently associated with a low risk of adverse oral events. However, our results suggest that adverse oral events also have a significant impact on outpatients receiving cancer drug therapies, highlighting their need for oral management by medical professionals.

Previous studies have reported that the incidence of dysgeusia with taxane chemotherapy is 55.0% [14]. Our results showed that taxane-based regimens had the highest incidence of dysgeusia. Taxane-based drug therapies affect neurosensory perception [15], and, therefore, the cause of dysgeusia was considered to be neuropathy induced by taxane-based drug therapies. Our study supports the notion that taxane-based regimens are strongly associated with dysgeusia. Given the significant impact of dysgeusia on patients’ nutritional status, quality of life, and treatment adherence [14,16,17,18], further research should focus on identifying risk factors and potential interventions.

According to the 2020 edition of the guidelines for managing mucositis during cancer drug therapy, anticancer drugs such as anthracycline-based, 5FU-based, and methotrexate regimens are associated with a higher incidence of oral mucositis [1]. Our results showed no significant differences in the incidence of oral mucositis among the cancer drug therapies. This discrepancy may be due to differences in study populations, as most previous studies focused on inpatients, while our study targeted outpatients. Additionally, the self-reported nature of our survey may have contributed to underreporting or variability in symptom perception. However, the incidence of oral mucositis was >25% for all regimens, excluding ICIs. Oral mucositis increases the risk of systemic complications, such as malnutrition, cachexia, and systemic infections [19,20]. Furthermore, oral mucositis negatively affects treatment outcomes and medical costs [21]. Therefore, the incidence of oral mucositis should be considered even in outpatient settings.

Our study showed the highest incidence of xerostomia when 5FU-based regimens were used. A previous study reported that patients receiving CPA, epirubicin, methotrexate, and 5FU experienced a higher incidence of xerostomia (64.0%) [22]. These findings suggest that xerostomia is a common adverse event in patients treated with 5FU-based regimens. Differences in the incidence rates between studies may be attributed to variations in assessment methods. Our study relied on self-reported patient evaluations, whereas the previous study used the UKU Side Effect Rating Scale. This difference in assessment approaches may have contributed to the variation in reported incidence rates. 5FU-based regimens are widely used to treat many solid tumors. They are also used in outpatient chemotherapy. Xerostomia can cause oral discomfort and functional impairment [23]. It results in a reduction in the physiological effects of saliva, including antibacterial activity, mucosal protection, and assistance with taste. Therefore, xerostomia may exacerbate dysgeusia and oral mucositis [24,25,26].

Our results indicate that oral adverse events related to cancer drug therapy significantly affect outpatients’ quality of life. More than a quarter of the patients required oral adverse event management by well-trained dentists and dental hygienists, highlighting the essential role of multidisciplinary collaboration in cancer care. Given the large number of outpatients undergoing cancer drug therapy, establishing an efficient screening and referral system for timely and appropriate oral management is critical to improving patient outcomes [27].

Our study has several limitations. First, the results were based on subjective patient-reported questionnaire surveys and did not reflect evaluations by medical professionals. Discrepancies may occur between patient and medical professional evaluations [28,29,30]. Self-reported data may lead to over- or underestimation. This study did not incorporate objective clinical assessments, such as standardized grading scales, dental examinations, or microbiological tests, limiting the accuracy of oral adverse event evaluation. Future studies should incorporate both patient-reported and clinician-reported outcomes to improve assessment reliability and develop better management strategies. Second, because the survey in this study was based on patient-reported questionnaires, the communication abilities of the patients and the absence of opinions from non-responders may have influenced the results. Third, although drug therapies containing anthracyclines or anti-EGFR antibodies are associated with a higher incidence of oral mucositis, only a limited number of patients were included in this study. Thus, assessment of the necessity for professional medical intervention in outpatients treated with these drug therapies was difficult. We performed multivariate analysis to reduce these biases. Fourth, because the survey was conducted at a single hospital, there was a potential bias in patient background and cancer drug therapy, leading to possible discrepancies. We plan to focus on 5FU-, taxane-, and anthracycline-based regimens and use more established adverse event assessment methods, such as PRO-CTCAE and CTCAE, for validation in future studies. Finally, our study had a cross-sectional design; therefore, the sample size was small. However, to maximize the number of participants, we requested consent from all eligible individuals.

## 5. Conclusions

Our results indicate that oral adverse events negatively affect the quality of life of outpatients receiving cancer drug therapy. More than a quarter of outpatients receiving cancer drug therapy needed oral management by well-trained dentists and dental hygienists. In particular, patients receiving 5FU-, taxane-, and anthracycline-based regimens need oral management by medical professionals.

Future studies must incorporate standardized evaluation methods such as PRO-CTCAE and CTCAE and collect data from a broader patient population through multicenter collaborative research. Adopting a longitudinal study design will be possible investigation of the time course of oral adverse events, as well as the impact of appropriate oral care on improving quality of life and maintaining treatment continuity. These approaches will contribute to the development of more effective and practical interventions, ultimately enhancing the quality of care for cancer patients.

## Figures and Tables

**Figure 1 cancers-17-00641-f001:**
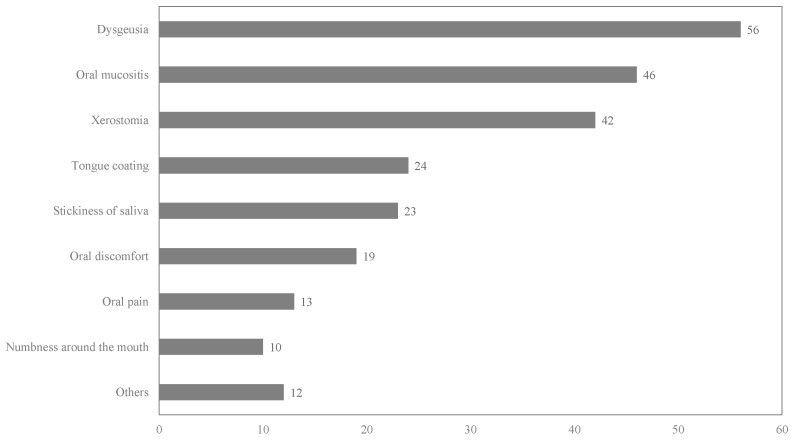
Details of the adverse events. This figure shows the number of patients with oral adverse events, as determined using the questionnaire. The survey allowed for multiple answers.

**Table 1 cancers-17-00641-t001:** Items on the patient evaluation questionnaire used in this study.

Q 1.	What type of oral adverse events do you have?
	1. Mucositis. 2. Tongue coating. 3. Dysgeusia. 4. Xerostomia. 5. Stickiness of saliva. 6. Numbness around the mouth. 7. Oral pain. 8. Oral discomfort. 8. None.
Q 2.	Do you want to improve the oral adverse events?
	1. Yes. 2. No.
Q 3.	Do the oral symptoms cause problems in your quality of life?
	1. Yes. 2. No.
Q 4.	What issues have become a problem in your quality of life?
	1. Reduced food intake. 2. Increased food intake. 3. Drinking too much. 4. Drinking not much. 5. Difficulty swallowing. 6. Sleeplessness. 7. Impaired speech. 8. None.
Q 5.	Do you want medication or a treatment method for oral adverse events?
	1. Yes. 2. No.
Q 6.	Do you want dentist intervention to improve your oral adverse events?
	1. Yes. 2. No
Q 7.	How severe are the oral adverse events compared with non-oral adverse events?
	1. More severe. 2. similar. 3. Not as severe.

**Table 2 cancers-17-00641-t002:** Patient characteristics (n = 216).

	Patient Characteristics	n	%
Sex	Male	108	50.0%
	Female	108	50.0%
Age group (years old)	AYA (18–39)	8	3.7%
	Middle-aged (40–69)	122	56.5%
	Old-aged (≥70)	86	39.8%
Cancer type	Breast cancer	52	24.1%
	Lung cancer	45	20.8%
	Gastrointestinal cancer	37	17.1%
	Liver, biliary tract, pancreatic cancer	18	8.3%
	Urological cancer	17	7.9%
	Gynecological cancer	14	6.5%
	Hematological cancer	12	5.6%
	Head and neck cancer	7	3.2%
	Soft tissue sarcoma	3	1.4%
	Others	11	5.1%
Cancer drug therapy	ICIs (Atezolizumab, Durvalumab, Pembrolizumab, Nivolumab, Avelumab)	67	31.0%
	Taxane-based (DTX, PTX, nab-PTX, CBZ)	49	22.7%
	5FU-based (5FU, TS-1, Cape)	33	15.3%
	Anthracycline-based	8	3.7%
	CDDP-based	4	1.9%
	EGFR antibodies (Cet, Pani)	3	1.4%
	Others	52	24.1%

AYA, Adolescent and young adult; ICIs, Immune checkpoint inhibitors; DTX, Docetaxel; PTX, Paclitaxel; nab-PTX, nab-Paclitaxel; CBZ, Cabazitaxel acetonate; Cape, Capecitabine; CDDP, Cisplatin; EGFR, Epidermal growth factor receptor; Cet, Cetuximab; Pani, Panitumumab.

**Table 3 cancers-17-00641-t003:** The relationship between patient characteristics and oral adverse events.

			With	Without	Multivariate Analysis		
		n	N (%)	N (%)	OR	95% CI	*p*-Value
		216	127 (58.8)	89 (41.2)			
Sex	Male	109	54 (49.5)	55 (50.5)			
	Female	107	73 (68.2)	34 (31.8)	1.14	0.48–2.70	0.76
Age group (years)	AYA (18–39)	8	4 (50.0)	4 (50.0)			
	Middle-aged (40–69)	122	77 (63.1)	45 (36.9)	3.21	0.59–17.70	0.18
	Older (≥70)	86	46 (53.5)	40 (46.5)	2.7	0.45–16.10	0.28
Cancer type	Others	11	4 (36.4)	7 (63.6)			
	Breast cancer	51	38 (74.5)	13 (25.5)	6.38	1.05–38.90	0.04 *
	Lung cancer	46	18 (39.1)	28 (60.9)	1.45	0.31–6.73	0.64
	Gastrointestinal cancer	37	27 (73.0)	10 (27.0)	2.79	0.42–18.40	0.29
	Liver, biliary tract, pancreatic cancer	18	12 (66.7)	6 (33.3)	1.42	0.21–9.70	0.72
	Urological cancer	17	6 (35.3)	11 (64.7)	1.29	0.22–7.62	0.78
	Gynecological cancer	14	9 (64.3)	5 (35.7)	6	0.82–43.90	0.08
	Hematological cancer	12	7 (58.3)	5 (41.7)	5.61	0.76–41.70	0.09
	Head and neck cancer	7	5 (71.4)	2 (28.6)	5.25	0.57–48.10	0.14
	Soft tissue sarcoma	3	1 (33.3)	2 (66.7)	0.9	0.05–17.30	0.95
Cancer drug therapy	Others	51	26 (51.0)	25 (49.0)			
	5FU-based (5FU, S1, Cape)	33	27 (81.8)	6 (18.2)	9.37	1.74–50.40	<0.01 *
	Taxane-based (DTX, PTX, nab-PTX, CBZ)	49	37 (75.5)	12 (24.5)	5.63	1.91–16.60	<0.01 *
	Anthracycline-based	8	6 (75.0)	2 (25.0)	1.95	0.35–10.90	0.45
	ICIs (Atezolizumab, Durvalumab, Pembrolizumab, Nivolumab, Avelumab)	68	27 (39.7)	41 (60.3)	1.69	0.55–5.19	0.36
	EGFR antibodies (Cet, Pani)	3	1 (33.3)	2 (66.7)	0.84	0.05–14.50	0.91
	CDDP	4	3 (75.0)	1 (25.0)	11.1	0.80–156.00	0.07

* Significant association (*p*-value  <  0.05). OR: odds ratio. CI: confidence interval.

**Table 4 cancers-17-00641-t004:** The relationship between cancer drug therapies and major oral adverse events.

			With	Without	Multivariate Analysis		
		n	N (%)	N (%)	OR	95% CI	*p*-Value
Dysgeusia		216	56 (25.9)	160 (74.1)			
	Others	51	9 (17.6)	42 (82.4)			
	5FU-based (5FU, S1, Cape)	33	12 (36.4)	21 (63.6)	1.36	0.23–8.19	0.74
	Taxane-based (DTX, PTX, nab-PTX, CBZ)	49	22 (44.9)	27 (55.1)	4.76	1.61–14.10	<0.01 *
	Anthracycline-based	8	4 (50.0)	4 (50.0)	3.50	0.71–17.30	0.13
	ICIs (Atezolizumab, Durvalumab, Pembrolizumab, Nivolumab, Avelumab)	68	7 (10.3)	61 (89.7)	0.74	0.19–2.89	0.66
	EGFR antibodies (Cet, Pani)	3	1 (33.3)	2 (66.7)	1.19	0.06–24.60	0.91
	CDDP	4	1 (25.0)	3 (75.0)	3.15	0.21–47.90	0.41
Oral mucositis		216	46 (21.3)	170 (78.7)			
	Others	51	13(25.5)	38 (74.5)			
	5FU-based (5FU, S1, Cape)	33	13 (39.4)	20 (60.6)	2.70	0.44–16.50	0.28
	Taxane-based (DTX, PTX, nab-PTX, CBZ)	49	14 (28.6)	35 (71.4)	1.85	0.65–5.25	0.25
	Anthracycline-based	8	2 (25.0)	6 (75.0)	0.74	0.13–4.21	0.73
	ICIs (Atezolizumab, Durvalumab, Pembrolizumab, Nivolumab, Avelumab)	68	4 (5.9)	64 (94.1)	0.45	0.10–2.00	0.29
	EGFR antibodies (Cet, Pani)	3	0 (0.0)	3 (100.0)	7.68 × 10^−8^	0-Inf	0.99
	CDDP	4	0 (0.0)	4 (100.0)	1.50 × 10^−7^	0-Inf	0.99
Xerostomia		216	42 (19.4)	174 (80.6)			
	Others	51	6 (11.8)	45 (88.2)			
	5FU-based (5FU, S1, Cape)	33	11 (33.3)	22 (66.7)	16.50	1.74–157.00	0.01 *
	Taxane-based (DTX, PTX, nab-PTX, CBZ)	49	11 (22.4)	38 (77.6)	2.65	0.80–8.79	0.11
	Anthracycline-based	8	2 (25.0)	6 (75.0)	2.05	0.32–13.10	0.45
	ICIs (Atezolizumab, Durvalumab, Pembrolizumab, Nivolumab, Avelumab)	68	11 (16.2)	57 (83.8)	2.29	0.55–9.53	0.25
	EGFR antibodies (Cet, Pani)	3	0 (0.0)	3 (100.0)	8.49 × 10^−7^	0-Inf	1.00
	CDDP	4	1 (25.0)	3 (75.0)	7.63	0.39–148.00	0.18

* Significant association (*p*-value < 0.05). OR: odds ratio. CI: confidence interval.

**Table 5 cancers-17-00641-t005:** Responses to the following question: Do you require management of oral adverse events by medical professionals?

		Response		
	Total	Yes		No
Item	N (%)	N (%)		N (%)
Hope for improvement of oral adverse events	115 (90.6)	68 (53.5)		47 (37.0)
Impact on their quality of life	113 (89.0)	44 (34.6)		69 (54.3)
Hope for professional oral care (Well-Trained Dentist and Dental Hygienist)	102 (80.3)	34 (26.8)		68 (53.5)
		more severe	similar	not as severe
Comparison of the severity of oral adverse events and other adverse events	113 (89.0)	5 (3.9)	27 (21.3)	81 (63.8)

## Data Availability

The datasets used and/or analyzed in the current study are available from the corresponding author upon reasonable request.

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
