# Peer review of "A Cross-Sectional Survey of Oral Adverse Events and Oral Management Needs in Outpatients Receiving Cancer Drug Therapy"

_cancers, 2025, doi:10.3390/cancers17040641_

Round 1

Reviewer 1 Report

Comments and Suggestions for Authors

      The objective of this study was to clarify the frequency and severity of oral adverse  events in outpatients receiving cancer drug therapy and to clarify patients' needs for oral adverse event management by dental professionals. The findings have several important  clinical implications.

The topic is interesting and well-written

This is a descriptive study with several gaps  methodology , it should delve deeper into the general mechanisms of action of the drugs to provide better context for the problem and include a hypothesis. The methodology is incomplete and requires additional data.

 ·         First, more than half of the outpatients receiving cancer drug therapy  had oral adverse events.

·         Second, more than a quarter of the patients who experienced oral adverse events felt that these symptoms were as severe or even more severe than other  non-oral adverse events. Third, more than a quarter of the patients who experienced oral adverse events wanted oral management through professional intervention by well-trained dentists and dental hygienists.

·         These results indicate that oral adverse events have  a significant negative impact on patients receiving outpatient cancer drug therapy.

The article should include more detailed information Cancer drug therapy about the dose of the drug administered and the duration of the treatment. Additionally, it is noted that the survey focuses exclusively on symptoms but does not address other important aspects such as side effects. It would be advisable to include data related to potential side effects, such as, periodontitis ,Gingival bleeding, the risk of cavities, and the incidence of oral infections, candidiasis..... Improving these points would provide a more comprehensive and enriched understanding of the treatment's impact on patients.

 the results were based on subjective patient-reported questionnaire surveys and did not reflect evaluations by medical professionals

the discussion should be improved.

Conclusions OK  

Author Response

Thank you for your time and efforts in helping us improve our manuscript and your helpful comments which were crucial in improving the scientific value of our paper.

- This is a descriptive study with several gaps  methodology , it should delve deeper into the general mechanisms of action of the drugs to provide better context for the problem and include a hypothesis. The methodology is incomplete and requires additional data.

 Thank you for pointing this out. We agree with your comment and modified the methodology as follows,”

Line85-90

In addition, we reviewed patient medical records to collect detailed information on sex, age, cancer type, and cancer drug therapy. Specifically, we recorded whether patients received 5-fluorouracil(5FU)-based (e.g., 5-fluorouracil, capecitabine), tax-anes-based (e.g., paclitaxel (PTX), docetaxel (DTX)), anthracycline-based (e.g., doxorubicin (DXR), epirubicin), or other regimens. This classification was based on the Clinical Guidance of Management for Mucositis, which categorizes agents with a high risk of inducing oral mucositis.

- The article should include more detailed information Cancer drug therapy about the dose of the drug administered and the duration of the treatment.

Thank you for pointing this out. We agree with your comment and modified the results as follows,”

Line112-126

ICIs (e.g., nivolumab (480mg/body, day1, every 28 days), pembrolizumab (400mg/body, day1, every 42 days)), taxanes-based regimens (e.g., cetuximab + PTX (250mg/m2 and 80mg/㎡, day1 every 7 days), DTX (60-75mg/ m2, day1, every 21 days), DTX + cyclophosphamide(CPA) (75mg/m2  and 600mg/m2, day1, every 21 days), PTX (80mg/m2, day1, every 7days), carboplatin + nab-PTX (AUC6 day1 and 100mg/m2, day1,8,15, every 21 days), nab-PTX + gemcitabine(GEM) (125mg/m2 and 1000mg/m2, day1,8,15, every 28 days)), 5FU-based regimens (e.g., 5FU + oxaliplatin(L-OHP) (400/2400 mg/m2 and 85mg/m2, day1, every 14 days), 5FU + irinotecan(CPT-11)(400/2400mg/m2 and 150mg/m2, day1, every 14 days), L-OHP + capecitabine (130mg/m2, day1 and 2000mg/m2, day1-14, every 21 days)), anthracycline-based regimens (e.g., DXR+CPA (60mg/m2 and 600mg/m2, day1, every 21 days), rituximab + DXR + vincristine + CPA + prednisolone (375mg/m2,50mg/m2,1.4mg/m2,750mg/m2, day1 and 60mg/m2 day1-5,every 21 days)), cisplatin (CDDP)-based regimens (e.g., CDDP + GEM (25mg/m2 and 1000mg/m2, day1,8, every 21 days) ) and monoclonal antibody drugs targeting epidermal growth factor receptor (EGFR) ( e.g., panitumumab (6mg/kg, day1, every 14 days).

- Additionally, it is noted that the survey focuses exclusively on symptoms but does not address other important aspects such as side effects. It would be advisable to include data related to potential side effects, such as, periodontitis ,Gingival bleeding, the risk of cavities, and the incidence of oral infections, candidiasis..... Improving these points would provide a more comprehensive and enriched understanding of the treatment's impact on patients.

Thank you for pointing this out. We agree with your comment and added the discussion as follows,”

Line234-236

This study did not incorporate objective clinical assessments, such as standardized grading scales, dental examinations, or microbiological tests, limiting the accuracy of oral adverse event evaluation.

- the results were based on subjective patient-reported questionnaire surveys and did not reflect evaluations by medical professionals

the discussion should be improved.

Thank you for pointing this out. We agree with your comment and modified the discussion as follows,”

Line231-238

First, the results were based on subjective patient-reported questionnaire surveys and did not reflect evaluations by medical professionals. Discrepancies may occur between patient and medical professional evaluations. Self-reported data may lead to over- or underestimation. This study did not incorporate objective clinical assessments, such as standardized grading scales, dental examinations, or microbiological tests, limiting the accuracy of oral adverse event evaluation. Future studies should incorporate both patient-reported and clinician-reported outcomes to improve assessment reliability and develop better management strategies.

Reviewer 2 Report

Comments and Suggestions for Authors

I really enjoyed reading the manuscript. It is very interesting to see the perception of patients and not only that of professionals. However, I found the introduction to be quite brief, and the authors should consider expanding it to provide more context.

Additionally, the discussion feels somewhat limited, as the authors do not sufficiently contrast their findings with those of other studies. Strengthening this section with a more in-depth comparison would improve the manuscript.

Author Response

Thank you for your time and efforts in helping us improve our manuscript and your helpful comments which were crucial in improving the scientific value of our paper.

- However, I found the introduction to be quite brief, and the authors should consider expanding it to provide more context.

 Thank you for pointing this out. We agree with your comment. Therefore, we have made significant changes to introduction. Please check them.

- Additionally, the discussion feels somewhat limited, as the authors do not sufficiently contrast their findings with those of other studies. Strengthening this section with a more in-depth comparison would improve the manuscript.

Thank you for pointing this out. We agree with your comment. Therefore, we have made significant changes to discussion. Please check them.

Reviewer 3 Report

Comments and Suggestions for Authors

Authors report on a "Cross-sectional survey of oral adverse events and oral management needs in outpatients receiving cancer drug therapy."

I do not understand this sentence (lines 30-32):

"Our results suggest that oral adverse events have a significant impact on outpatients receiving cancer drug therapy."

This is a well-known fact; it cannot be the result/conclusion of a study!

If the statement refers to the frequency and severity of adverse events, then it is acceptable. However, the authors should modify it to:

"Our results suggest that cancer drug therapy, frequency and severity of oral adverse events, has a significant impact on the outpatients' quality of life/daily life."

The authors' survey is important, so the results should be presented in a way that makes them useful for the readers of Cancers.

Every oncologist and dentist knows that cancer drug therapy can cause severe oral side effects. A study like this cannot establish that fact -it is common knowledge.

Why is the institution where the research was conducted not listed in the affiliations?

None of the authors work at that Center Outpatient Cancer Chemotherapy Center of Niigata…..). Did no one from that Center participate in the study? Did any clinicians (oncologist, dentist) contribute to the research?

It should be emphasized more clearly that the aim of the study was to compare the differences between the adverse events of specific drug therapies.

Would it be better to use "quality of life" instead of "daily life"?

At line 200, I would write "needed" instead of "wanted".

I do not believe "dental intervention" was required, but rather oral adverse event management performed by a well-trained onco-dentist (e.g., for mucositis, candidiasis). Please ensure that these distinctions are made accurately throughout the entire article.

An important, valuable paper cannot conclude with vague statements such as:

"Our results suggest… future studies should… furthermore…"

It should present clear and concrete statements!

With the above modifications, clarifications, and corrections, I recommend the article for publication.

Author Response

Thank you for your time and efforts in helping us improve our manuscript and your helpful comments which were crucial in improving the scientific value of our paper.

- I do not understand this sentence (lines 30-32):

"Our results suggest that oral adverse events have a significant impact on outpatients receiving cancer drug therapy."

This is a well-known fact; it cannot be the result/conclusion of a study!

If the statement refers to the frequency and severity of adverse events, then it is acceptable. However, the authors should modify it to:

"Our results suggest that cancer drug therapy, frequency and severity of oral adverse events, has a significant impact on the outpatients' quality of life/daily life."

The authors' survey is important, so the results should be presented in a way that makes them useful for the readers of Cancers.

Every oncologist and dentist knows that cancer drug therapy can cause severe oral side effects. A study like this cannot establish that fact -it is common knowledge.

Thank you for pointing this out. We agree with your comment and modified the sentence as follows,

Our results suggest that cancer drug therapy, and the frequency and severity of oral adverse events, have a significant impact on the outpatients’ quality of life.

- Why is the institution where the research was conducted not listed in the affiliations?

Thank you for pointing this out. We agree with your comment and modified as follows,”

Yuki Sakai 1,2*, Kouji Katsura 3, Masaaki Kotake 1,2 and Akira Toyama 1

1 Department of Pharmacy, Niigata University Medical and Dental Hospital

2 Department of Outpatient Cancer Chemotherapy Center, Niigata University Medical and Dental Hospital

3 Department of Oral Radiology, Niigata University Medical and Dental Hospital ;

- It should be emphasized more clearly that the aim of the study was to compare the differences between the adverse events of specific drug therapies.

Thank you for pointing this out. We agree with your comment and modified the introduction as follows,”

Line63-67

This study aimed to bridge this gap by examining the reported data on the frequency and severity of oral adverse events in outpatients receiving cancer drug therapy. Further objectives were to clarify patient needs for the oral management, and to identify drugs that affect oral adverse events, facilitating the development of tailored management strategies for this growing patient population.

- Would it be better to use "quality of life" instead of "daily life"?

Thank you for pointing this out. We agree with your comment and modified "daily life" to "quality of life".

- At line 200, I would write "needed" instead of "wanted".

Thank you for pointing this out. We agree with your comment and modified "wanted" to "needed".

- I do not believe "dental intervention" was required, but rather oral adverse event management performed by a well-trained onco-dentist (e.g., for mucositis, candidiasis). Please ensure that these distinctions are made accurately throughout the entire article.

Thank you for pointing this out. We agree with your comment.

We ensure that these distinctions are made accurately throughout the entire article.

- An important, valuable paper cannot conclude with vague statements such as:

"Our results suggest… future studies should… furthermore…"

It should present clear and concrete statements!

Thank you for pointing this out. Therefore, we have made significant changes to conclusions. Please check them.

Round 2

Reviewer 1 Report

Comments and Suggestions for Authors

The authors have responded well to the feedback, and the manuscript has been noticeably improved.